# Assembly status transition offers an avenue for activity modulation of a supramolecular enzyme

Yao Chen[1†], Weiya Xu[1†], Shuwei Yu[2†], Kang Ni[3], Guangbiao She[2], Xiaodong Ye[3], Qiong Xing[4*], Jian Zhao[2*], Chengdong Huang[1*]

[1]Ministry of Education Key Laboratory for Membrane-less Organelles & Cellular Dynamics, Hefei National Laboratory for Physical Sciences at the Microscale, School of Life Sciences, Division of Life Sciences and Medicine, University of Science and Technology of China, Hefei, China; [2]State Key Laboratory of Tea Plant Biology and Utilization, College of Tea and Food Science and Technology, Anhui Agricultural University, Hefei, China; [3]Hefei National Laboratory for Physical Sciences at the Microscale, Department of Chemical Physics, University of Science and Technology of China, Hefei, China; [4]State Key Laboratory of Biocatalysis and Enzyme Engineering, Hubei Collaborative Innovation Center for Green Transformation of Bio-Resources, Hubei Key Laboratory of Industrial Biotechnology, School of Life Sciences, Hubei University, Wuhan, China

*For correspondence:
qiongxingnmr@hubu.edu.cn
(QX);
jzhao2@qq.com (JZ);
huangcd@ustc.edu.cn (CH)

†These authors contributed
equally to this work

Reviewing Editor: Philip A Cole,
Harvard Medical School, United
States

**Abstract** Nature has evolved many supramolecular proteins assembled in certain, sometimes even seemingly oversophisticated, morphological manners. The rationale behind such evolutionary efforts is often poorly understood. Here, we provide atomic-resolution insights into how the dynamic building of a structurally complex enzyme with higher order symmetry offers amenability to intricate regulation. We have established the functional coupling between enzymatic activity and protein morphological states of glutamine synthetase (GS), an old multi-subunit enzyme essential for cellular nitrogen metabolism. Cryo-EM structure determination of GS in both the catalytically active and inactive assembly states allows us to reveal an unanticipated self-assembly-induced disorder-order transition paradigm, in which the remote interactions between two subcomplex entities significantly rigidify the otherwise structurally fluctuating active sites, thereby regulating activity. We further show in vivo evidences that how the enzyme morphology transitions could be modulated by cellular factors on demand. Collectively, our data present an example of how assembly status transition offers an avenue for activity modulation, and sharpens our mechanistic understanding of the complex functional and regulatory properties of supramolecular enzymes.

## Editor's evaluation

This study provides deep structural insights into how an important enzyme, glutamine synthase, which exists in at least two large multimeric complexes, can be catalytically regulated in different assemblies. The paper combines high resolution cryoEM, enzymology, and cellular studies to provide a plausible model for how the intricate structural changes regulate activity.

## Introduction

Recent studies have evidenced that only a small portion of proteins function in isolation in cells, whereas the majority is assembled into complexes through protein-protein interactions with identical or different protein subunit(s) (**Marsh et al., 2013**). The rationale behind such an evolutionary selection has been the subject of considerable speculation; proposals for the advantages associated with a multimeric-units complex instead of a long single polypeptide chain include better error control in synthesis, greater coding and folding efficiency, and possibility of allosteric regulation (**Goodsell and Olson, 2000**). Morphologically speaking, many protein complexes especially homomeric ones adopt a symmetric spatial arrangement, either cyclic ($C_{n (n>1)}$) or dihedral ($D_{n (n>1)}$) symmetry, characterized by a rotational symmetry or two orthogonal symmetry axes, respectively. In contrast to the cyclic complexes which evolve in one step (e.g. C1→C5), evolution of dihedral complexes takes place in multiple steps (e.g. C1→C5→D5) (**Levy et al., 2008**), and adds another layer of structural complexity. Intriguingly, pioneering studies have revealed many supramolecular enzymes organized in dihedral symmetry, with subcomplex entities in cyclic symmetry holding, at least outwardly, multiple integral active sites. Thus, a fundamental question arises here is that why nature builds these protein complexes with a seemingly oversophisticated quaternary design, if the subcomplexes alone possess complete elements for action? In other words, is the extra assembly step, e.g. C5→D5, is a futile evolutionary effort for these supramolecular protein complexes?

One such an example is glutamine synthetases (GSs) (EC 6.3.1.2), one of the most ancient functioning enzymes in existence and a central enzyme in nitrogen metabolism of all living organisms, catalyzing the formation of glutamine by condensation of glutamate with ammonia using ATP as an energy source (**Eisenberg et al., 2000**; **Stadtman and Ginsburg, 1974**). Three classes of GS enzymes have been identified in different organisms, namely, GSI, GSII, and GSIII. Decades of studies have established a striking notion that all three classes of GS enzymes, despite of dramatic differences in amino acid sequences and protein sizes, share quaternary geometry in dihedral symmetry assembled with two oligomeric rings (**Brown et al., 1994**; **van Rooyen et al., 2011**; **Almassy et al., 1986**; **Gill and Eisenberg, 2001**; **Gill et al., 2002**; **Unno et al., 2006**; **Torreira et al., 2014**; **Krajewski et al., 2008**). Considering that the active sites of GS are located at the clefts formed between two neighboring protomers within the same ring and distal to the ring-ring interface, each isolated GS subcomplex ring holds multiple integral catalytic sites (**Eisenberg et al., 2000**; **Betti et al., 2012**). The functional demand for this evolutionary conservation, the quaternary organization of GS with dihedral symmetry, remains elusive.

Here, we sought out to explore the functional link between the oligomeric conformation and catalysis activity, and mechanistically justify the seemingly oversophisticated assembly design in this supramolecular enzyme. Our results unveil a previously uncharacterized structural disorder-order transition along with activity regulation mechanism induced by assembly status transition of GS, and present an example that how a particular quaternary geometry selectively defines the oligomer dynamics congruent with required activities. We further show in vivo evidence how this regulatory machinery is elegantly utilized by the cell to meet the ever-changing metabolic needs. The functional implications of these findings are discussed.

## Results

### Two highly conserved GSIIs demonstrate distinct quaternary structure organization propensities

With the aim of clarifying the functional role of dihedral symmetry in GS functions, we first carried out a quest for GSs that share a high degree of sequence conservation, but demonstrate distinct quaternary structural assembly properties. We made use of the weak ring-ring interaction of GSII, a prominent structural difference between the type I and type II GSs (**Unno et al., 2006**; **Torreira et al., 2014**; **Krajewski et al., 2008**), built model structures for the candidate GSIIs based on the crystal structure of the maize GSII (pdb code:2D3B) and analyzed the amino acid variations in the context of model structures. Primary structure analysis reveals GSIIs from the plants of *Camellia sinensis* (CsGSIb) and *Glycine max* (GmGSβ2) share an overall very high sequence homology (~90 % identical and ~97 % conserved) and absolutely conserved substrate-binding and catalytic sites (**Figure 1a**), with, however, a significant portion of amino acid variations clustered at the interface between two pentamer rings

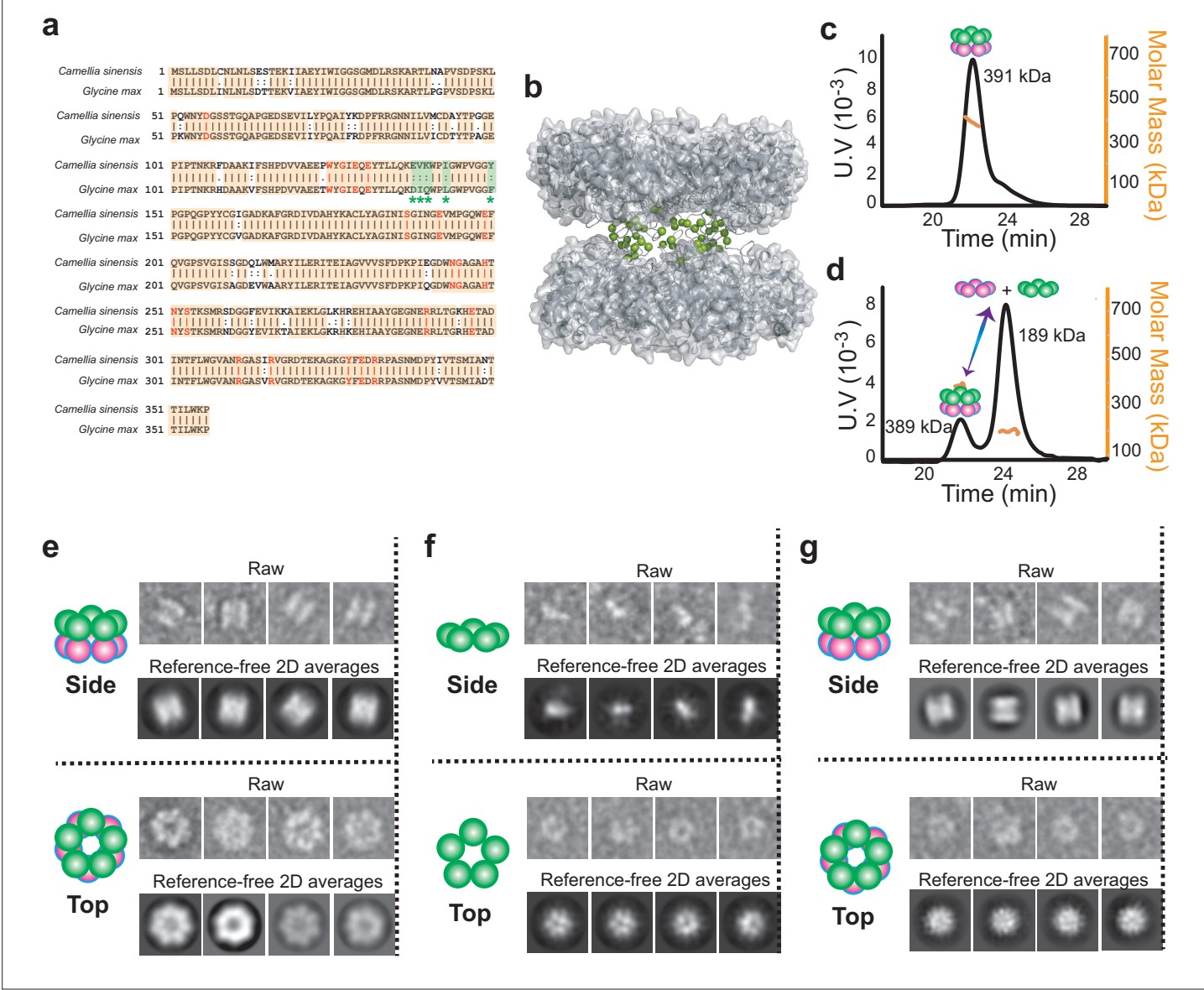

**Figure 1.** Quaternary assembly property comparison of GSIIs from *Camellia sinensis* (CsGSIb) and *Glycine max* (GmGSβ2). (**a**) Amino acid sequence alignment of CsGSIb and GmGSβ2 reveals very high level of conservation. Identical amino acids are shown with orange boxes, while the residues involved in substrate-binding and catalysis are shown in red. Amino acids variations located at the pentamer ring-ring interface are highlighted in green boxes with the symbol of *. (**b**) Model structure built based on the crystal structure of a maize GSII (GmGSβ2, pdb code 2D3B). The amino acid variations between CsGSIb and GmGSβ2 that are located at the ring-ring interface are highlighted as spheres in green. (**c–d**) SEC-MALS analysis of GmGSβ2 (**c**) and CsGSIb (**d**). (**e–g**) Quaternary assembly analysis of GmGSβ2 and CsGSIb using negative-stain electron microscopy. Left: A schematic representation of the averages is shown for clarity; Right upper: Examples of single raw images; Right lower: reference-free two-dimensional class averages. GmGSβ2 adopts a homogenous double-ringed structure (**e**), while the CsGSIb demonstrates a mixture of two major classes of particles: isolated pentamer ring (**f**) and double-ringed structure (**g**).

The online version of this article includes the following figure supplement(s) for figure 1:

**Figure supplement 1.** CsGs1b exhibits a pentamer-decamer dynamic equilibrium in solution.

(*Figure 1a and b*). We then recombinantly expressed both CsGSIb and GmGSβ2 in *E. coli* and purified these two GSII homologs. To assess the oligomerization status, we performed size-exclusion chromatography (SEC) coupled to both multi-angle light scattering (MALS) and quasi-elastic light scattering (QELS). MALS analysis shows the GSII from *Glycine max* being largely a homogeneous decamer in solution (*Figure 1c*). In contrast, under the same condition the majority fraction (~82%)

of CsGSIb adopts a pentameric configuration, along with a minor fraction (~18%) being decameric (*Figure 1d*). We further show that CsGSIb exists in pentamer-decamer dynamic equilibrium in solution and a mixture of electrostatic and hydrophobic interactions is responsible for attaching of two pentameric rings; whereas substrates or ligands show no appreciable effect on the decamer-forming properties (*Figure 1—figure supplement 1*).

We further employed negative-stain electron microscopy (EM) to directly visualize the distinct ring-ring packing propensities for CsGSIb and GmGSβ2. Two-dimensional (2D) class averages revealed that GmGSβ2 forms homogeneous, double stacked-ring shaped particles (*Figure 1e*), in line with the decameric organization pattern previously reported for other GSII species (*Unno et al., 2006*; *Torreira et al., 2014*; *Krajewski et al., 2008*). In contrast, CsGSIb adopted a mixture of two quaternary structural modes: detached pentamers (*Figure 1f*) and decamers composed of two stacked pentamer rings (*Figure 1g*), consistent with the above MALS analysis result (*Figure 1d*).

Analytical ultracentrifugation (AUC) was carried out to quantitively assess the thermodynamic parameters of CsGSIb pentamer-decamer transition, which demonstrated two species for CsGSIb in solution with molecular weights of 191 kDa and 395 kDa (*Figure 2a*), corresponding to the pentameric and decameric configurations, respectively. Sedimentation profiles at various protein concentrations were analyzed and a global analysis of the data yielded a pentamer-pentamer dissociation constant ($K_d$) of 0.27 ± 0.06 µM at room temperature. It is noteworthy that the dissociation constant within the sub-micromolar range allows CsGSIb to predominantly exist as isolated pentamers under the concentration assayed for enzymatic activities, laying a solid foundation for further probing the functional role of the decamer formation in modulating the enzyme activity of GS.

## GSIIs in different assembly states demonstrate distinct enzymatic activities

We next sought to compare the glutamine synthesis activity of these two highly conserved GSIIs. As shown in *Figure 2b* and *Figure 2—figure supplement 1a*, when supplied with ammonium chloride, the stable decamer-adopting GmGSβ2 demonstrated significant GS activities. Further steady-state kinetic measurements yielded turnover numbers ($k_{cat}$) and Michaelis constants ($K_m$) of ~12.8 min$^{-1}$ and ~44 µM, respectively (*Figure 2c*). In sharp contrast, CsGSIb only demonstrated basal activity (*Figure 2b* and *Figure 2—figure supplement 1a*), consistent with previous observations that the isolated single-ringed GSII species is nonfunctional (*Llorca et al., 2006*; *Mäck, 1998*). These observations raised the question as to whether the drastic disparity in catalytic activities could be attributed to their distinct propensities for formation of a double-ringed architecture. If the above proposal holds true, a positive concentration-dependent cooperation of enzyme activity would be expected as increase in the concentration of CsGSIb favors decamer assembly (*Figure 1—figure supplement 1a*). Indeed, five times increase in the concentration of CsGSIb assayed (from 1 µM to 5 µM) resulted in ~30 folds increase in activity, that is ~6 folds activity increase per unit of enzyme, displaying a significant concentration-dependent simulation effect (*Figure 2—figure supplement 1b*).

## Catalytic switching of CsGSIb through oligomeric states interconversion

To further validate the above proposal, we performed mutagenesis to CsGSIb, aiming to convert its unstable pentamer-decamer equilibrium state to a stable decamer and then evaluate the impact on catalytic activity. According to the amino acid sequence of GmGSβ2, three CsGSIb mutants, that is EVK138DIQ, I143L and Y150F, respectively, were prepared as the wild-type (WT) protein. SEC-MALS measurements revealed both mutations of I143L and EVK138DIQ led to a drastic shift in the oligomerization equilibrium towards decamer assembly (*Figure 2d and f*), causing significant increase in the distribution of decameric state from ~18 % in the WT-CsGSIb (*Figure 1d*) to >95% and ~ 72%, respectively. These observations confirmed that both mutations introduced at the interface, albeit largely conservative, dramatically fortified the decamer edifice assembly. In contrast, substitution of the tyrosine at the residue 150 with a phenylalanine showed no appreciable change in its quaternary organization mode (*Figure 2—figure supplement 2a*), suggesting no critical role for the residue Y150 in maintaining two CsGSIb pentameric ring subcomplexes attached. Further AUC analysis revealed replacement of I143 or EVK at residues 138–140 with leucine or DIQ yielded ring-ring disassociation constants of ~0.01 or ~ 0.04 µM, respectively (*Figure 2e and g*), that is, ~ 27 or ~ 7 folds of

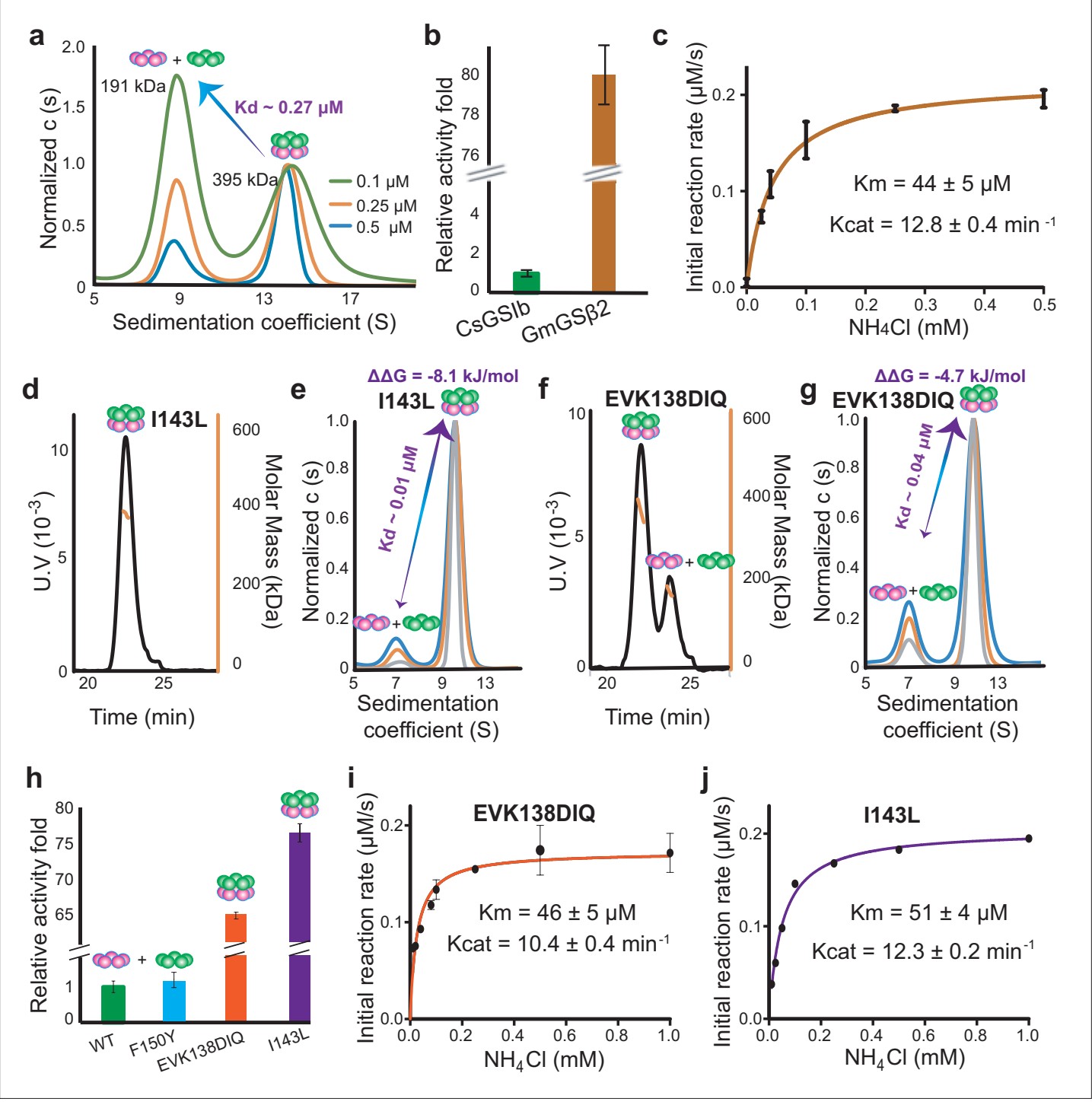

**Figure 2.** The enzymatic activities of CsGSIb is dependent on its quaternary assembly status. (**a**) Application of analytical ultracentrifugation (AUC) to assess the pentamer-decamer dissociation constant of CsGSIb. Experiments were performed at room temperature with three different sample concentrations shown as monomer concentration. A global fit of sedimentation distribution profiles yielded a dissociation constant of 0.27 μM. (**b**) Enzymatic activity comparison of GmGSβ2 with CsGSIb. Reactions were performed for 30 min at 37 °C in presence of 1 μM (monomer) enzyme and saturated amounts of substrates. (**c**) Steady-state kinetic analysis of GmGSβ2. Assay conditions were the same as that in (**b**), except the concentrations of NH$_4$Cl were varied. (**d-g**) Mutation effects on the quaternary assembly property of CsGSIb evaluated using SEC-MALS (**d and f**) and AUC (**e and g**). The corresponding mutants are labeled in the figures. (**h**) Activity comparison of the wild type CsGSIb with its mutants as labeled. Reactions were performed in the same condition as (**b**). (**i-j**) Steady-state kinetic analysis of CsGSIb mutants of EVK138DIQ (**i**) and I143L (**j**). Reaction conditions were same as in (**c**). All enzyme assays were repeated at least three times and data were shown as means ± s.d.

*Figure 2 continued on next page*

*Figure 2 continued*

The online version of this article includes the following figure supplement(s) for figure 2:

**Figure supplement 1.** Enzymatic characterizatioin of GmGSβ2 and CsGSIb.

**Figure supplement 2.** Mutation effects of F150Y on the quaternary assembly property and enzyme activity of CsGSIb.

---

increase in the ring-ring binding affinity compared with that of WT-CsGSIb. The Gibbs energy changes upon mutation (ΔΔG) for I143L and EVK138DIQ were calculated to be –8.1 kJ/mol and –4.7 kJ/mol, respectively.

We next set out to investigate the resulting impacts on their enzymatic activities. As shown in *Figure 2h–j*, both mutations of I143L and EVK138DIQ caused dramatic increase in catalytic activity of ~76 and ~ 64 folds, respectively. As all mutated amino acids are distal to either the catalytic site or substrate binding regions with distances > 20 Å and hence are unlikely to be directly involved in catalytic reaction, we infer that the stimulations of the enzymatic activity of CsGSIb upon residue perturbations are induced by remote contacts between two pentamer rings. As expected, the mutation of Y150F, which did not alter pentamer-decamer equilibrium of CsGSIb, showed no noticeable change in enzymatic activity (*Figure 2—figure supplement 2b*).

## The disorder-order transition induced by quaternary assembly triggers enzymatic activation of GSII

To elucidate the detailed mechanism of how the interactions between two GSII pentameric rings remotely trigger enzymatic activity, we next employed single-particle cryo-EM imaging technique and first determined the structures of GmGSβ2 decamer, as well as that of the CsGSIb that adopts decameric configuration (thereafter named as CsGSIb^Dec). 3D classifications of 104,717 and 43,876 particles for GmGSβ2 and CsGSIb^Dec, respectively, revealed that both molecules were arranged in D5 symmetry with two pentameric rings stacked in a head-to-head manner (*Figure 3a–b*), a strikingly conserved structural feature for type II GS species (*Unno et al., 2006*; *Torreira et al., 2014*; *Krajewski et al., 2008*). Refinement of the GmGSβ2 and CsGSIb^Dec structures yielded maps with an average resolution of 2.9 Å and 3.3 Å, respectively (*Figure 3—figure supplements 1–3*), with literally identical dimensions of 115 Å x 115 Å x 95 Å. As expected from the very high conservation in amino acid sequence (*Figure 1a*), decamer structures of GmGSβ2 and CsGSIb^Dec are strikingly similar to each other, as well as to that of the GSII of maize (pdb accession number 2D3A), as highlighted by the root-mean-square deviation (r.m.s.d.) of 0.66–0.80 Å for 328–352 aligned C$_\alpha$ atoms. The active sites of GmGSβ2, CsGSIb^Dec are structurally highly similar to each other, as well as to that of the maize GSII (*Figure 3c*), suggesting the catalysis mechanisms of these plant GSIIs are essentially identical. Indeed, structural alignments of GSs reveals a conservative catalytic site geometry across a wide range of species (*Supplementary file 1*).

The overall buried inter-ring surfaces for both GmGSβ2 and CsGSIb^Dec amount to ~2000 Å², that is approximately only 400 Å² per individual monomer–monomer interaction. This highlights the weakness of the inter-ring contacts, characteristic of type II GS. Indeed, in both structures the inter-ring contacts are established by only a limited number of hydrophobic and polar interactions provided by the residues 136–141 and 146–152 segments of each of the intervening subunits (*Figure 3—figure supplement 4*), which behave as two gear teeth (thereafter named as tooth-1 and tooth-2, respectively) interlocking the two pentameric rings (*Figure 3d–f*). This observation is in line with the above result that mutations to tooth-1 resulted in drastic change in oligomeric states behavior (*Figure 2d–g*), and the mixed nature of the inter-ring interactions is consistent with the MALS analysis result of CsGSIb under various buffer conditions (*Figure 1—figure supplement 1*). Intriguingly, the local structure of the teeth regions that mediate inter-ring interactions remains largely the same in GmGSβ2 and CsGSIb^Dec (*Figure 3d–e*), suggesting the dramatically different propensities of GmGSβ2 and CsGSIb for decamer formation are due to the nature of the amino acids involved in inter-ring contacts, rather than the structure. Although the residue of I143 is not directly involved in inter-ring contact, we argue that its replacement with leucine may stabilize the conformation of tooth-1 via its interaction with the residue of L134, thus playing an important role in stabilizing decamer architecture (*Figure 2d and e*).

In order to elucidate the mechanism of how the pentameric CsGSIb (thereafter named as CsGSIb^Pen) demonstrates distinct enzymatic properties than CsGSIb^Dec (*Figure 2h*), we next determined the

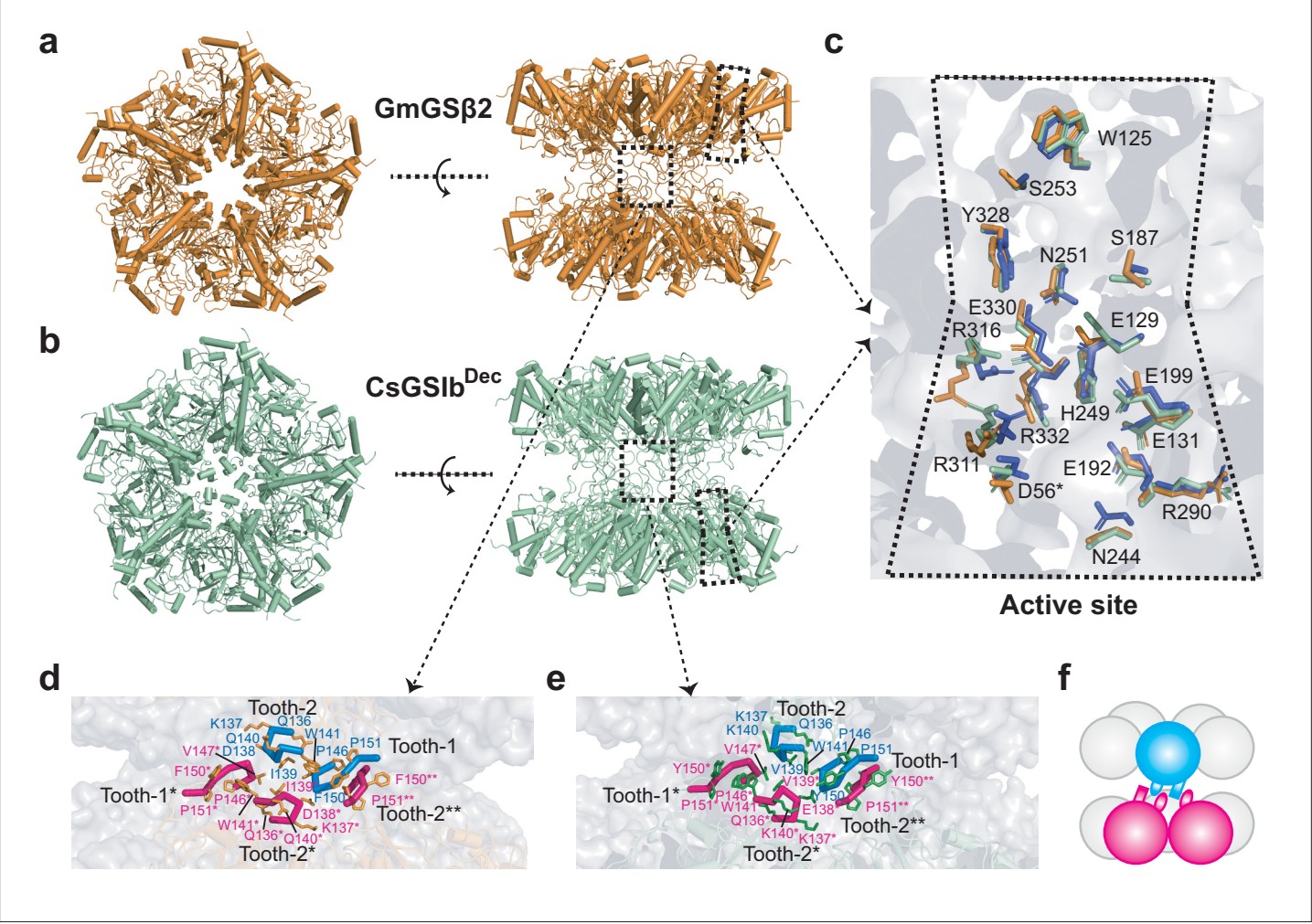

**Figure 3.** Overall structures, catalytic centers and ring-ring interfaces of GmGSβ2 and CsGSIb$^{Dec}$. (**a-b**) Overall double-ringed structures of GmGSβ (**a**) and CsGSIb$^{Dec}$ (**b**). Left: top-view; Right: Side-view. (**c**) Superimposed structures of the catalytic centers of GmGSβ2 (orange), CsGSIb$^{Dec}$ (green) and GSII from maize (purple). High structure similarities in catalytic sites suggest the catalytic mechanism for these three GSII species are essentially identical. (**d-e**) The detailed ring-ring interaction interfaces between GmGSβ2 (**d**) and CsGSIb$^{Dec}$ (**e**). The interactions between two pentameric rings are primarily mediated by two regions, namely, the tooth-1 and tooth-2, respectively. (**f**) A schematic representation of the two GSII pentameric rings interlocked by tooth-1 and tooth-2 for clarity.

The online version of this article includes the following figure supplement(s) for figure 3:

**Figure supplement 1.** Cryo-EM analysis of GmGSβ2.

**Figure supplement 2.** Cryo-EM analysis of CsGSIb.

**Figure supplement 3.** Validation of cryo-EM structures.

**Figure supplement 4.** Detailed analysis of interactions between two GSII pentameric rings using the program of Ligplot.

**Figure supplement 5.** Swinging motion of CsGSIb rings.

structure of CsGSIb$^{Pen}$. Lowering the sample concentration, which favored pentamer dissociation (*Figure 1—figure supplement 1a*), allowed us to obtain sufficient number of CsGSIb$^{Pen}$ particles (*Figure 3—figure supplements 2 and 3*), which, in turn, enabled us to solve the cryo-EM structure of the inactive single-ringed GSII for the first time. Interestingly, we observed additional class averages in which the two masses of density attributed to the pentameric rings are no longer parallel (*Figure 3—figure supplement 5*). These nonparallel ring particles may reflect intermediate assembly stages in the formation/disruption of the enzyme decamer, again confirming the flexibility of the inter-ring interactions of the type II GS.

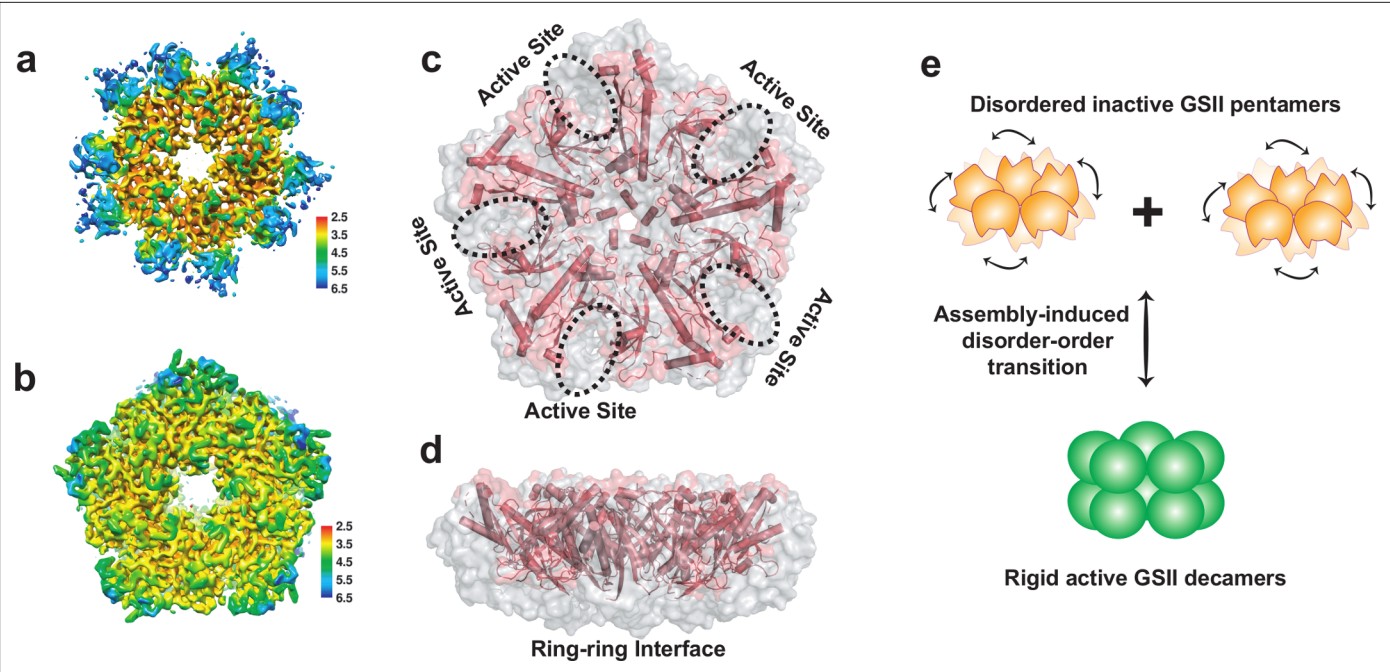

**Figure 4.** Cryo-EM structure of CsGSIb[Pen] features high conformational flexibility. For simplicity, only one out three CsGSIb[Pen] cryo-EM structures are presented here, and all three structures are shown in *Figure 4—figure supplement 1*. (**a**) Local resolution of the density map of CsGSIb[Pen] indicates a decreased resolution near the edges of the pentamer ring. (**b**) Local resolution of the density map of CsGSIb[Dec] map. The conformational flexibility is apparent when the missing density at the rim of CsGSIb[Pen], which yields a largely decagram-shaped map (**a**), is compared to the intact density of CsGSIb[Dec] that displays a pentagon-shaped map (**b**). (**c–d**) Superimposed of Cryo-EM structure of CsGSIb[Pen] (pink) with that of CsGSIb[Dec] (grey surface). (**c**): Top-view; (**d**): Side-view. The results reveal two major regions being highly disordered: the rim region including the catalytic center, and ring-ring interface. (**e**) Proposed activation mechanism model of CsGSIb.

The online version of this article includes the following figure supplement(s) for figure 4:

**Figure supplement 1.** Three cryo-EM structures of CsGSIb pentamers in isolation.

**Figure supplement 2.** Thermal shift assays.

Unexpected, CsGSIb[Pen] exhibited high conformational heterogeneity and 3D particles classification generated three similar structures with the r.m.s.d. ranging from 0.6 to 0.8 Å, differing in a few peripheral regions (*Figure 4—figure supplement 1*). The most striking difference between CsGSIb[Pen] and CsGSIb[Dec] is that several regions are missing in the electron density map of all three classifications of CsGSIb[Pen] particles, with only 229–255 out of 356 residues electron densities in presence. As a result, CsGSIb[Pen] demonstrated a decagram-shaped density map with a few regions missing at the rim (*Figure 4a* and *Figure 4—figure supplement 1a*), in sharp contrast to a pentagon-shaped map yielded by the CsGSIb[Dec] particles (*Figure 4b*). For the 229–255 residues that show clear density in CsGSIb[Pen] particles, the conformation of each subunit in three CsGSIb[Pen] EM structures, as well as the arrangement pattern, closely resembles that of the CsGSIb[Dec], as evidenced by r.m.s.d. in the range of 0.7–1.0 Å (*Figure 4—figure supplement 1b-d*). The density-missing regions include the segments around residues of 110–117, 140–166, and 260–334, among which, the fragment around residues 260–334 is a major component making up an integral catalytic site (*Figure 4c*), while the segment of residues 140–166 comprising of the two gear teeth is responsible for ring-ring interaction (*Figures 3d–f , and 4d*). As electron density missing often reflects the conformational heterogeneity arising from internal motions *Armache and Cheng, 2019*, these observation strongly suggest that the conformation of CsGSIb[Pen] active site is highly dynamic, contrasting sharply to the conformationally largely homogeneous CsGSIb[Dec]. In support this, thermal shift assays show the melting temperature ($T_m$) of wild-type CsGSIb is significantly lower than that of its mutants of I143L or EVK138DIQ, indicating of structural instability for the pentameric GSII (*Figure 4—figure supplement 2*). We therefore conclude that the dramatically difference in the dynamic property of catalytic sites accounts for the distinct activities of pentameric and decameric CsGSIb.

Taken together, our cryo-EM structures allow us to propose a disorder-order transition mechanism of how the GSII activity is regulated by changes in oligomeric state: (1) The active sites within isolated CsGSIb[Pen] rings are highly disordered and the unstable catalytic environments render it catalytically inactive; (2) Upon stacking of two pentameric rings and formation of a decamer, the signals of interactions mediated by the gear teeth of each intervening subunit are propagated to the active sites and induce folding, which, in turn, unlock the catalytic potential of GSII (*Figure 4e*).

## Activation of the GSII by the 14-3-3 scaffold protein

Having mechanistically established the functional coupling between GSII activity and its quaternary assembly status, we next asked whether there exist cellular factors that may regulate the CsGSIb activity, potentially via favoring its decamer assembly. 14-3-3 proteins are an important family of scaffold proteins that bind and regulate many key proteins involved in diverse intracellular processes in all eukaryotic organisms (*Tzivion et al., 2001*; *Chevalier et al., 2009*; *Obsilova and Obsil, 2020*). In particular, self-dimerization of 14-3-3 proteins, which induces dimerization of their clients, plays a key role in its functional scaffolding and subsequent activity regulation (*Tzivion et al., 2001*; *Chevalier et al., 2009*; *Kondo et al., 2019*). It has been reported that 14-3-3 proteins act as an activator of GSs in various plants (*Finnemann and Schjoerring, 2000*; *Pozuelo et al., 2001*; *Lima et al., 2006*; *Riedel et al., 2001*), although the detailed activation mechanism remains unclear. Based on these findings, here we tentatively provide the missing link in mechanistically assigning the role 14-3-3 proteins play in regulating GS activity: One protomer of the 14-3-3 protein recognizes one phosphorylated GSII pentamer, and its self-dimerization brings two pentamer rings in close proximity and therefore promotes decamer assembly, which, in turn, switches on the GS activity via rigidification of the catalytic sites (*Figure 5a*). One prerequisite for this proposal is that, for the GS species whose activities being 14-3-3 protein-dependent, they must have an intrinsically weak decamer-forming propensity that is to be overcome by 14-3-3. In support of this, the GS from *Medicago truncatula*, whose activity is simulated upon binding to 14-3-3 protein (*Lima et al., 2006*), has been shown to exhibit a dynamic pentamer-decamer transition (*Torreira et al., 2014*), similar to the CsGSIb presented here (*Figure 1d*). Moreover, it has been shown that only the higher order complex of tobacco GS-2 that is bound to 14-3-3 is catalytically active (*Riedel et al., 2001*).

To further support the above proposal, we then explored whether the activity of the weak-decamer forming CsGSIb could also be regulated by 14-3-3 scaffold protein. Homology search against tea plant genome revealed several candidate tea plant Cs14-3-3 proteins (*Figure 5—figure supplement 1*). Analysis of their coding genes' expression patterns in tea plant tissues and in nitrogen assimilation or metabolism-related processes allowed us to identify *Cs14-3-3-1a* and *Cs14-3-3-1b* genes that displayed expression patterns highly similar to *CsGSI* genes (*Figure 5a* and *Figure 5—figure supplement 2*). Moreover, the expression levels of *Cs14-3-3-1a* and *Cs14-3-3-1b* genes were regulated upon changes in the availability of ammonia (*Figure 5b and c*), the substrate of GS, suggesting both *Cs14-3-3-1a* and *Cs14-3-3-1b* are physiologically related to GS. We then examined the in vivo interactions between Cs14-3-3-1a and CsGSIb using the bimolecular fluorescence complementation (BiFC) technique, which is based on complementation between two non-fluorescent fragments of a fluorescent protein when they are brought together by interactions between proteins fused to each fragment *Kerppola, 2006*. Cs14-3-3-1a or CsGSIb was fused in frame with N-terminal half of a yellow florescence protein (NYFP) or C-terminal half of a yellow florescence protein (CYFP), respectively, and expressed in tobacco leaf epidermal cells alone or in various combinations, such as CsGSIb [CYFP] alone or together with Cs14-3-3-1a[NYFP]. As expected, Cs14-3-3-1a and 1b could self-dimerize or form heterodimers in plant cells (*Figure 5e*), consistent with 14-3-3 scaffold proteins adopting a dimeric structure (*Tzivion et al., 2001*; *Chevalier et al., 2009*; *Kondo et al., 2019*). Importantly, formation of the fluorescent complex clearly demonstrated the interaction of Cs14-3-3-1a with CsGSIb (*Figure 5f*). In order to further establish the functional relevance, we performed the RNA interference (RNAi) technique to knock down the transcript level of *Cs14-3-3-1a* gene in hairy roots of chimerical transgenic tea seedlings (*Figure 5h*), and evaluated the impact on GS activity by measuring the contents of GS catalysis product, glutamine. We show that, along with the reduction in *Cs14-3-3-1a* transcript level, the glutamine contents (*Figure 5i*), as well as the crude enzyme activity (*Figure 5j*), were drastically reduced, indicating the 14-3-3 protein in *Camellia sinensis* functions as an activator molecule of CsGSIb. Further work is needed to elucidate the detailed mechanism of how *Cs14-3-3-1a* recognizes phosphorylated CsGSIb.

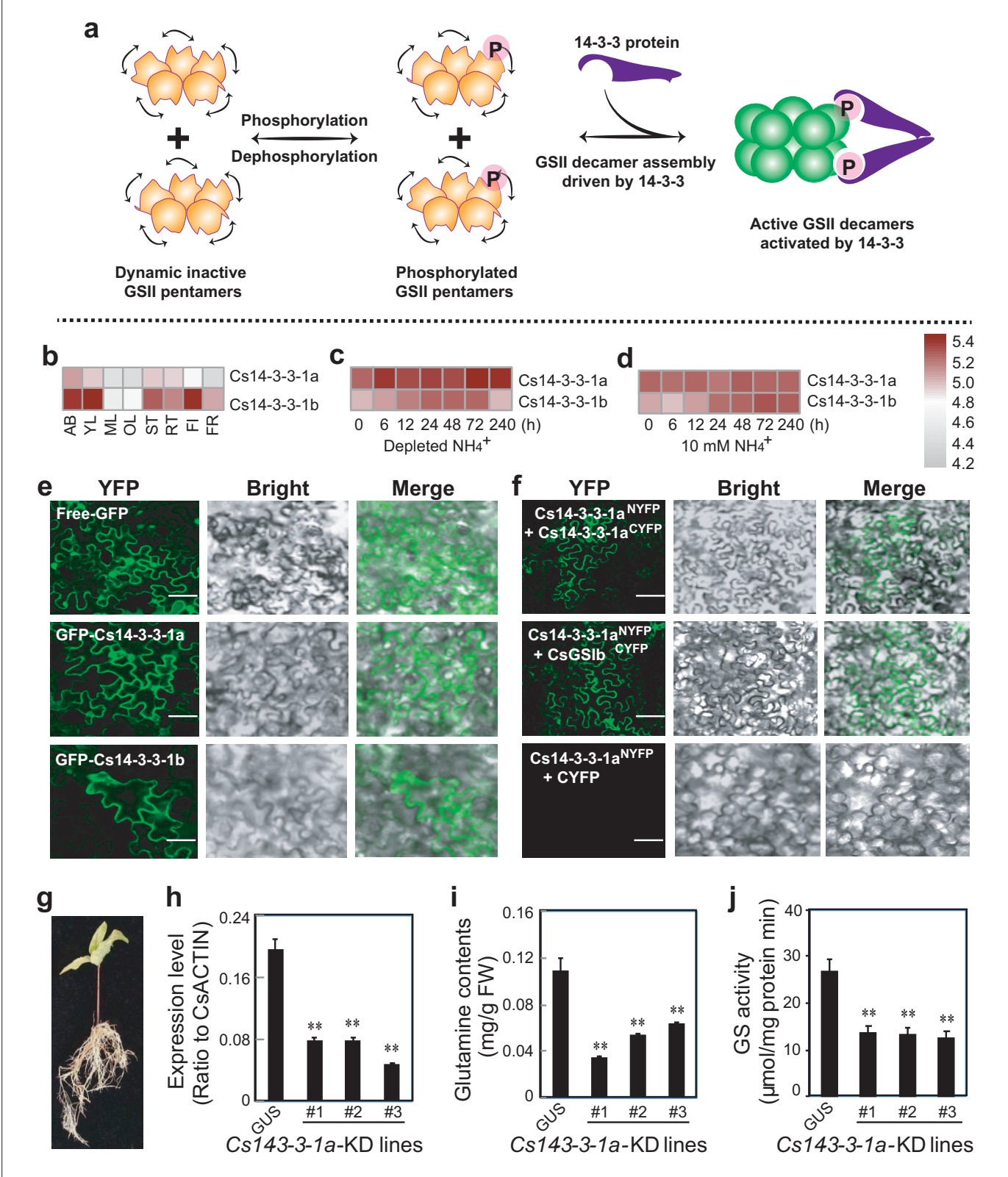

**Figure 5.** Activation of the CsGSIb by the 14-3-3 scaffold protein. (**a**) Proposed working model of how 14-3-3 protein modulates the activity of GSIIs. P in pink sphere denotes the post-translational modification of phosphorylation. (**b**) Expression patterns of *Cs14-3-3-1a* and *Cs14-3-3-1b* genes in various tea plant tissues. (**c**) Induction of Cs14-3-3-1a and *Cs14-3-3-1b* genes by depletion of $NH_4^+$ from culture medium. (**d**) Expression of *Cs14-3-3-1a* and *Cs14-3-3-1b* genes in tea plant roots fed with 10 mM $NH_4^+$. AB, apical buds of unopened leaves at the top of actively growing shoots; YL, first and second

*Figure 5 continued on next page*

*Figure 5 continued*

young leaves below the apical buds; ML, mature leaves geminated in the spring and harvested in the autumn; OL, old leaves at the bottom of tea tree plant; FL, flowers; FR, fruits of tea plants; ST, stem tissues at the 2nd and 3rd internodes; RT, roots. (**e**) Subcellular localization of *Cs14-3-3-1a* and *Cs14-3-3-1a* fusion proteins in tobacco leaf epidermal cells. bar = 50 μM. (**f**) BIFC assay of interaction among Cs14-3-3-1a, Cs14-3-3-1b and CsGSIIb in tobacco leaf cells. bar = 50 μM. (**g**) Generation of tea plant transgenic hairy roots with RNAi knockdown(KD) of *Cs14-3-3-1a* gene. (**h**) Expression of Cs14-3-3-1a in at least three tea plant transgenic hairy roots of *Cs14-3-3-1a-KD* compared with *GUS* roots. i, Glutamine contents in three tea plant transgenic hairy roots of *Cs14-3-3-1a-KD* compared with *GUS* roots. (**j**) GS activity in three tea plant transgenic hairy roots of *Cs14-3-3-1a-KD* compared with *GUS* roots. All experiments were conducted at least three three independent experiments. At least five transgenic hairy roots for *Cs14-3-3-1a* and *GUS* genes were examined. Data are expressed as means ± s.d. Differences in two-tailed comparisons between transgenic lines and *GUS* controls were analyzed, \*\*$p < 0.01$ in student's t-test. See Materials and methods for experimental details.

The online version of this article includes the following figure supplement(s) for figure 5:

**Figure supplement 1.** Phylogenetic analysis of Cs14-3-3s from Camellia sinensis genome as compared with 14-3-3s from *Arabidopsis* and rice.

**Figure supplement 2.** Expression of *Cs14-3-3* genes in eight tissues of *Camellia sinensis* plants.

## Discussion

Proper assembly of individual protein units into functional complexes is fundamental to nearly all biological processes. Comparing to the oligomer assembled in relatively simple cyclic symmetry that contains only interfaces of subunits related by the rotational symmetry, protein complexes organized in dihedral symmetry, an extra step of assembly during evolution, possess interfaces that are related by both the rotational symmetry and the perpendicular twofold axes. However, in many cases, the functional demand for such structural complexity remains poorly understood. Here, by using GS as a model system, we unveil a previously uncharacterized functionally regulatory code buried in a supramolecular protein complex with dihedral symmetry, and show how dynamic packing of protein subcomplexes could build an extra control for activity modulation.

### The disorder-order transition mechanism of GSII offers a robust and tunable regulatory machinery

Being a key enzyme implicated in many aspects of the complex matrix of nitrogen metabolism, GS must be strictly regulated. Decades of efforts have been applied to understand how GS is controlled at gene, transcript and protein levels (*Bernard and Habash, 2009*). Studies have demonstrated positive cooperativity of GS with regard to different substrates and cofactors, such as L-glutamate (*Montanini et al., 2003*) and metal cations (*Sakakibara et al., 1996*); and GS functions are regulated by multiple post-translational mechanisms including nitration, oxidative turnover and phosphorylation (*Seabra and Carvalho, 2015*), and by the 14-3-3 protein (*Finnemann and Schjoerring, 2000*; *Pozuelo et al., 2001*; *Lima et al., 2006*; *Riedel et al., 2001*). Our results reconcile with many of the above observations, and allow us to gain a more complete picture of how GS activity is regulated in cells by an exquisite machinery (*Figure 6*). While the protein turnovers machineries to adjust cellular enzyme level can always provide means to modulate pentamer-decamer transitions and thus deactivation-activation conversion of GSII, we argue that phosphorylation-dephosphorylation processes, coupled with 14-3-3 binding and subsequent activation, may enable a more efficient regulatory way. When sufficient reaction products are available demanding low glutamine synthesis activity, GSII is kept in the dephosphorylated state and exists as isolated inactive single-ringed pentamers. In the physiological context of high demand of glutamine, phosphorylation of GSII by certain kinase prompts 14-3-3 protein binding, and the intrinsic dimerization property of 14-3-3 recruits two GSII pentamer rings in close proximity and in doing so, result in a rapid transition of quaternary assembly from the pentamer to decamer, and eventually enzymatic activation. In this manner, the poised GSII pentamer ring itself acts as a positive effector and the functional ring-ring association offers a great advantage of immediate response to precisely meet the ever-changing metabolic needs, whereas the reversible assembly-disassembly behavior enables a tunable mode for activity modulation. Indeed, the dynamic association-disassociation of GSII subcomplexes, a prerequisite for this modulatory machinery, have been widely observed in various species including humans (*Denman and Wedler, 1984*), plants other than *Camellia sinensis* reported here (*Torreira et al., 2014*; *Llorca et al., 2006*) and fungi (*Mora, 1990*). Therefore, the activation mechanism shown here may represent a general regulatory machinery harnessed by many eukaryotes to ensure optimal utilization of nitrogen sources, and the

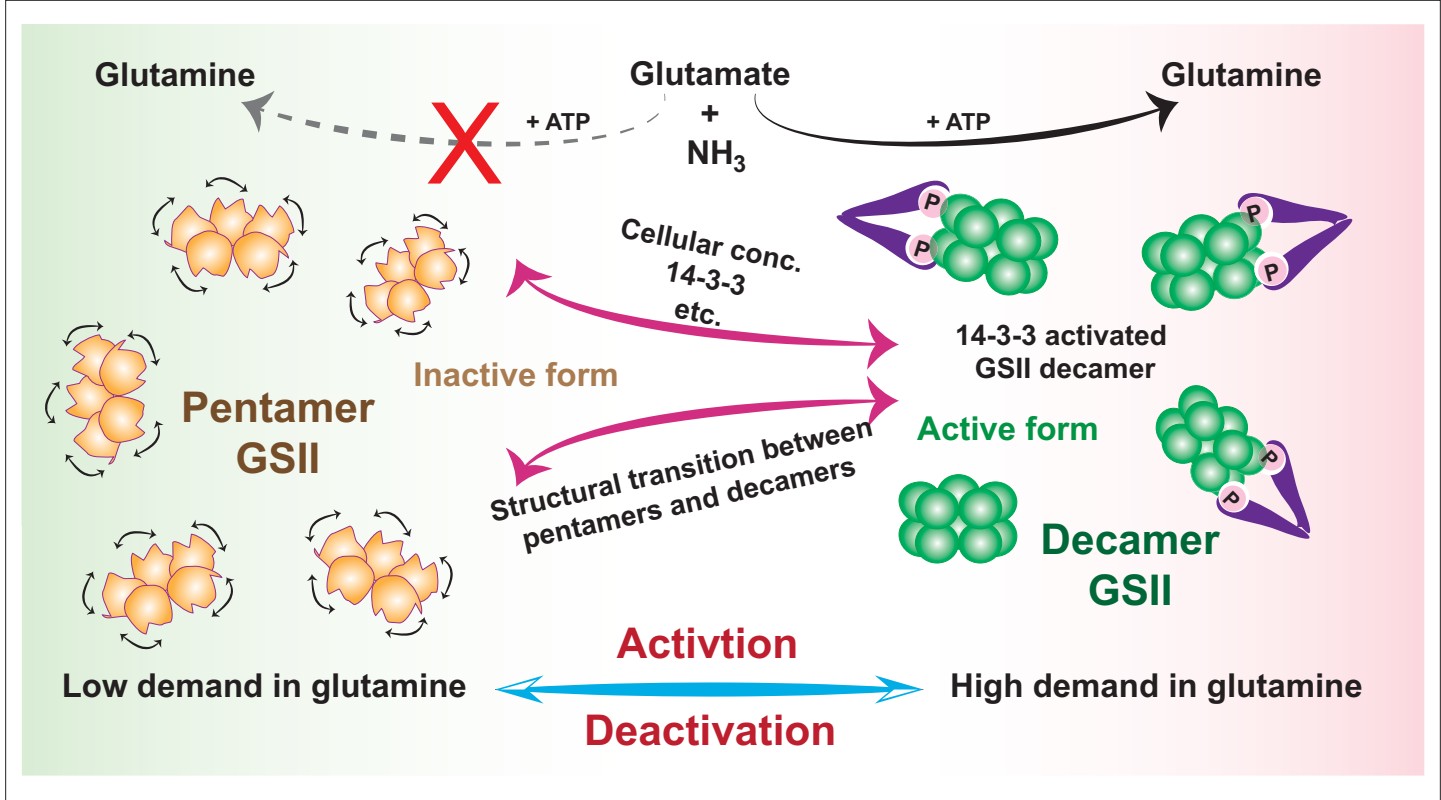

**Figure 6.** Schematic model of how activity of GSII is regulated by assembly states transition in cells to meet metabolic needs. In low demand of glutamine, GSII is kept in the dephosphorylated state and exists as isolated inactive pentamers (orange); In the physiological context of high demand of glutamine, phosphorylation and subsequent binding to 14-3-3 (or by increase in cellular concentration) trigger a rapid transition of quaternary assembly of GSII from the pentamer to decamer (green) and activates GSII.

infrastructure of fragile ring-ring contacts evolutionarily chosen by many eukaryotes offers a convenient and robust avenue for activity regulation.

## Practical implications

As a crucial enzyme to all living organisms, which is involved in all aspects of nitrogen metabolism, GS has emerged as an attractive target for drug design (*Mowbray et al., 2014*) and herbicidal compounds development, as well as a suitable intervention point for the improvement of crop yields (*Thomsen et al., 2014*). However, because the overall geometry of the active site is the most conserved structural element amongst GS enzymes (*Eisenberg et al., 2000*; *van Rooyen et al., 2011*; *Unno et al., 2006*), the traditional strategy of selective inhibition, which relies heavily on the subtle difference in the active sites from different species, has only achieved limited success. Thus, the regulatory mechanism discovered here will help guide the search for specific inhibitors of potential therapeutic interest. For example, inhibition of the GS in *Mycobacterium tuberculosis* has long been recognized as a novel antibiotic strategy to treat tuberculosis (*Harth and Horwitz, 2003*; *Gising et al., 2012*; *Kumari and Subbarao, 2020*). Our result opens new possibilities to develop chemicals to target the drugable ring-ring interface region and specifically interrupt the interactions between two GS subcomplexes in pathogens or unwanted plants to develop new types of herbicide. Moreover, although overexpression of GS has been investigated extensively for decades with the goal of improving crop nitrogen use efficiency, the outcome has not been consistent (*Thomsen et al., 2014*). The modulatory 'hot-spots' identified here, which mediate inter-ring communication and in turn stimulate GS activity, will guide engineering catalytically more powerful GSs for crops in which pentamer units only weakly associate, and thus increase plant nitrogen use efficiency and crop production.

# Materials and methods

## Key resources table

| Reagent type (species) or resource | Designation | Source or reference | Identifiers | Additional information |
|---|---|---|---|---|
| Gene (*Camellia sinensis*) | CsGSIb | Genbank | No. MK716208 | |
| Gene (*Glycine max*) | GmGSβ2 | Genbank | No. NM001255403 | |
| Strain, strain background (*Agrobacterium tumefaciens*) | EHA105 strain | C58 | | |
| Strain, strain background (*Agrobacterium rhizogenes*) | 15834 strain | ATCC | | |
| Strain, strain background (*Escherichia coli*) | Rosetta(DE3) | Sigma-Aldrich | 70,954 | Electrocompetent cells |
| Sequence-based reagent | CsGSII-1b(I143L)_FWD | This paper | PCR primers | GAAGTGAAGTGGCCGCTTGGTTGGCCTGTG |
| Sequence-based reagent | CsGSII-1b (I143L)_REV | This paper | PCR primers | CACAGGCCAACCAAGCGGCCACTTCACTTC |
| Sequence-based reagent | CsGSII-1b(Y150F)_FWD | This paper | PCR primers | CCTGTGGGAGGTTTTCCTGGACCACAG |
| Sequence-based reagent | CsGSII-1b (Y150F)_REV | This paper | PCR primers | CTGTGGTCCAGGAAAACCTCCCACAGG |
| Sequence-based reagent | CsGSII-1b (EVK138DIQ)_FWD | This paper | PCR primers | CACTTTGTTGCAGAAAGATATCCAGTGGCCGATTGGTTGGCC |
| Sequence-based reagent | CsGSII-1b (EVK138DIQ)_REV | This paper | PCR primers | GGCCAACCAATCGGCCACTGGATATCTTTCTGCAACAAAGTG |
| Sequence-based reagent | Forward primer for Cs14-3-3-1a-KD Knockdown | This paper | RNAi-PCR primers | GGGGACAAGTTTGTACAAAAAAGCAGGCTTCCGACGCGCGAAGAGAACGTGT |
| Sequence-based reagent | Reverse primer for Cs14-3-3-1a-KD Knockdown | This paper | RNAi-PCR primers | GGGGACCACTTTGTACAAGAAAGCTGGGTGGAGCAAGTTCTGCACTTGC |
| Sequence-based reagent | Forward primer for tagged fusion gene Cs14-3-3-1a-GFP | This paper | PCR primers | GGGGACAAGTTTGTACAAAAAAGCAGGCTTCATGGCGGTGGCACCATCGACG |
| Sequence-based reagent | Reverse primer for tagged fusion gene Cs14-3-3-1a-GFP | This paper | PCR primers | GGGGACCACTTTGTACAAGAAAGCTGGGTTCACTTCTGTTCTTCATCTTG |
| Sequence-based reagent | Forward primer for Cs14-3-3-1b-GFP-F | This paper | PCR primers | GGGGACAAGTTTGTACAAAAAAGCAGGCTTCATGGCGGTGGCAGCATCGGCA |
| Sequence-based reagent | Reverse primer for Cs14-3-3-1b-GFP | This paper | PCR primers | GGGGACCACTTTGTACAAGAAAGCTGGGTTCACTGCTTTTCATCATCGTG |
| Sequence-based reagent | Forward primer for Cs14-3-3-1a-CYFP-F | This paper | PCR primers | TATATTCTGCCCAAATTCGCGATGGCGGTGGCACCATCGACG |
| Sequence-based reagent | Reverse primer for Cs14-3-3-1a- CYFP-R | This paper | PCR primers | GTCGGCGAGCTGCACGCTGCCCTTCTGTTCTTCATCTTGCTT |
| Sequence-based reagent | Forward primer for Cs14-3-3-1a-NYFP-F | This paper | PCR primers | TATATTCTGCCCAAATTCGCGATGGCGGTGGCACCATCGACG |

*Continued on next page*

*Continued*

| Reagent type (species) or resource | Designation | Source or reference | Identifiers | Additional information |
|---|---|---|---|---|
| Sequence-based reagent | Reverse primer for Cs14-3-3-1a- NYFP-R | This paper | PCR primers | CTCCTCGCCCTTGCTCACCATCTT CTGTTCTTCATCTTGCTT |
| Sequence-based reagent | Forward primer for CsGSIb-CYFP-F | This paper | PCR primers | TATATTCTGCCCAAATTCGCGATG TCTCTTCTTTCCGATCTT |
| Sequence-based reagent | Reverse primer for CsGSIb-CYFP-R | This paper | PCR primers | GTCGGCGAGCTGCACGCTGCCC GGTTTCCAGAGGATGGT |
| Sequence-based reagent | Forward primer for CsGSIb-NYFP-F | This paper | PCR primers | TATATTCTGCCCAAATTCGCGA TGTCTCTTCTTTCCGATCTT |
| Sequence-based reagent | Reverse primer for CsGSIb-NYFP-R | This paper | PCR primers | CTCCTCGCCCTTGCTCACCAT CGGTTTCCAGAGGATGGT |
| Commercial assay or kit | ClonExpress II One Step Cloning Kit | Vazyme | C112-01 | |
| Chemical compound, drug | AMP-PNP | Sigma Aldrich | CAS: 25612-73-1 | |
| Chemical compound, drug | ATP | Sigma Aldrich | CAS: 34369-07-8 | |
| Chemical compound, drug | L-ascorbic acid | Sigma Aldrich | CAS: 50-81-7 | |
| Chemical compound, drug | ammonium molybdate tetrahydrate | Sigma Aldrich | CAS: 12054-85-2 | |
| Chemical compound, drug | sodium citrate tribasic dehydrate | Sigma Aldrich | CAS: 6132-04-3 | |
| Chemical compound, drug | Methyl Alcohol | Tedia | MS1922-801 | |
| Chemical compound, drug | Shigeki Konishi solution | SINOPHARM | | |
| Chemical compound, drug | Glutamine | Sigma Aldrich | G3126 | |
| Biological sample (*Camellia sinensis*) | Camellia sinensis var sinensis cultivar Shuchazao | Dechang seed seedling Co., Ltd | | |
| Software, algorithm | Cryosparc | DOI:10.1038/nmeth.4169 | Cryosparc v2 | |
| Software, algorithm | I-TASSER | DOI:10.1093/nar/gkv342 | I-TASSER server | |
| Software, algorithm | UCSF Chimera | doi:10.1002/jcc.20084 | UCSF Chimera 1.15 | https://www.cgl.ucsf.edu/chimera |
| Software, algorithm | COOT | doi:10.1107/ S0907444910007493 | | http://www2mrc-lmbcamach .uk/personal/pemsley/coot/ |
| Software, algorithm | Phenix | doi:10.1107/ S2059798318006551 | | https://www.phenix-online.org/ |
| Software, algorithm | PyMOL | PyMOL Molecular Graphics System, Schrodinger LLC | | https://www.pymol.org/ |

## Cloning, expression, and protein purification of GSs

The target genes encoding CsGSIb from *Camellia sinensis* (Genbank accession No. MK716208) and GmGSβ2 from *Glycine max* (Genbank accession No. NM001255403) were cloned into the pET-16b vector (Novagen) containing a His$_6$-tagged-MBP tag followed by a tobacco etch virus (TEV) protease cleavage site at the N-terminus. All constructs were transformed into *E. coli* Rosetta (DE3) cells, which were cultured in Luria-Bertani (LB) medium at 37 °C supplemented with ampicillin (100 μg/ml) and chloramphenicol (35 μg/ml) to an OD600 ~0.8. Cells were induced by the addition of

isopropyl-β-D-1- thiogalactopyranoside (IPTG) to the concentration of 0.3 mM, and incubated for additional 16 hr at 18 °C. Cells were harvested by centrifugation at 5000 g for 20 min and resuspended in lysis buffer (50 mM Tris-HCl, 500 mM NaCl, pH 8 and 1 mM PMSF).

Cells were subjected to a high-pressure homogenizer, named JN-Mini Pro Low-temperature Ultrahigh-pressure cell disrupter (JNBIO) and then centrifuged at 50,000 g for 30 min at 4°C. Proteins were initially purified using Ni Sepharose 6 Fast Flow resin (GE Healthcare). The protein tags were cleaved with His-tagged TEV-protease overnight at 4 °C while dialyzing against TEV cleavage buffer (50 mM Tris-HCl, 100 mM NaCl, 1 mM β-mercaptoethanol, pH 8). Cleaved sample was collected and run over Ni-NTA column to remove His-tagged TEV and protein tags. Flow-through was collected, concentrated and passed over Hiload 16/600 Superdex 200 column (GE Healthcare) in 50 mM Tris-HCl, pH 7.4, 100 mM NaCl, 0.5 mM $MgCl_2$ and 1.5 mM β-mercaptoethanol.

## Multi-angle light scattering (MALS) characterization

MALS was measured using a DAWN HELEOS-II system (Wyatt Technology Corporation) downstream of a GE liquid chromatography system connected to a Superdex 200 10/300 GL (GE Healthcare) gel filtration column. The running buffer for the protein samples contained 50 mM KPi (pH 7.0), 100 mM NaCl, 1 mM β-mercaptoethanol and 0.05 % $NaN_3$. The flow rate was set to 0.5 mL min$^{-1}$ with an injection volume of 200 μL, and the light scattering signal was collected at room temperature (~23 °C). The data were analyzed with ASTRA version 6.0.5 (Wyatt Technology Corporation).

## Glutamine synthetase activity assay

GS activity assay was performed described previously (*Gawronski and Benson, 2004*; *Masalkar and Roberts, 2015*) and reactions were performed for 30 min at 37 °C in 50 mM Tris-HCl, pH 7.4, 100 mM NaCl, 0.5 mM $MgCl_2$ and 1.5 mM β-mercaptoethanol. Enzymatic activity comparison was conducted with 1 μM (monomer) enzyme in the presence of 0.5 mM $NH_4HCl$, 2 mM L-glutamate, 0.5 mM ATP. Steady-state kinetic analysis was performed under the same conditions except with the variable concentration of ammonium chloride from 0.05 mM to 4 mM. Steady-state kinetic parameters were determined by double reciprocal Lineweaver-Burk plot for reactions that followed Michaelis-Menten kinetics. All experiments were repeated independently at least three times.

## Fluorescent dye-monitored thermal shift assays

Reactants containing 2 uM CsGSIb (monomer) and 1000-fold diluted Sypro Orange in 50 mM Tris-HCl, pH 7.4, 100 mM NaCl and 1.5 mM β-mercaptoethanol were performed using an iCycler thermocycler (Bio-Rad) as previous described (*Krajewski et al., 2008*). Briefly, CsGSIb in presence of various concentrations of the following ligands were tested, alone and in combination: 10 mM glutamate, 20 mM $MgCl_2$ and 1 mM ATP. The temperature of the reactions was increased from 20°C to 90 °C in increments of 0.2 °C/12 s, coincident with a fluorescent measurement at each step. The wavelengths for excitation and emission were set to 490 and 575 nm, respectively. Fluorescence changes were monitored simultaneously with a charge-coupled device (CCD) camera. To obtain the temperature midpoint for the protein unfolding transition, Tm, a Boltzmann model was used to fit the fluorescence imaging data obtained by the CCD detector using the curve-fitting software GraphPad Prism 7.0.

## Analytical ultracentrifugation (AUC)

Sedimentation velocity experiments were carried out with a Proteomelab XL-A analytical ultracentrifuge (Beckman Coulter, USA) using a four-hole An-60 Ti analytical rotor. An aliquot of 410 μL of buffer (50 mM Tris-HCl, pH 7.4, 100 mM NaCl and 1.5 mM β-mercaptoethanol) as the reference and 400 μL of protein solution (0.1/0.25/0.5 mg·mL$^{-1}$) were loaded into a double-sector cell. A centerpiece with a path length of 12 mm was used. The speed of rotor was 35,000 rpm. The operation temperature of rotor was 20°C. The time dependence of the absorbance at different radial positions was monitored at a wavelength of 280 nm by an UV–Vis absorbance detector, and the data were analyzed by the software SEDFIT (version 15.01b) using c(s) model to obtain the sedimentation coefficient distribution. Viscosity and density of the buffer solution were calculated by the Sednterp software.

## Single-particle cryo-electron microscopy data collection

Purified protein samples of CsGSIb (4 μL, 0.02 mg/mL) and GmGSβ2 (4 μL, 0.02 mg/mL) in 50 mM Tris-HCl, pH 7.4, 100 mM NaCl, and 1.5 mM β-mercaptoethanol were negatively stained with uranyl acetate 1 % (w/v) on carbon-film 400 mesh copper grids. Samples were imaged using a FEI T12 operated at 120 keV with a 3.236 Å pixel size, 68,000× nominal magnification, and defocus range about 1.5 μm. For cryo-EM, 3 μL of CsGSIb (0.1 mg/mL and 0.5 mg/mL) and GmGSβ2 (0.1 mg/mL) were added onto glow-discharged Quantifoil R1.2/1.3 100 holey-carbon Cu grids with a Vitrobot Mark IV (Thermo Fisher Scientific). The grids were blotted for 3.5 s at 8 °C with 100 % humidity, and then plunged frozen into liquid ethane cooled by liquid nitrogen. Cryo-grids were first screened on a FEI TF20 operated at 200 keV. Images of CsGSIb and GmGSβ2 were collected using Titan Krios G3i microscope (FEI) operated at 300 kV with a Gatan K2 Summit direct detection camera. Two datasets were acquired using the SerialEM in super-resolution mode with a nominal magnification of 29,000 x, yielding a pixel sizes of 0.505 Å with a total dose of 51 e/ $Å^2$. The defocus ranges were set from −1.6 μm to −2.3 μm.

## Cryo-electron microscopy image processing, 3D reconstruction, and analysis

All processing steps were performed using cryoSPARC (*Punjani et al., 2017*). A total of 4051 raw movie stacks acquired for CsGSIb and 1777 raw movie stacks for GmGSβ2 were subjected to patch motion correction and patch CTF estimation. An initial set of about 500 particles were manually picked to generate 2D templates for auto-picking. The auto-picked particles were extracted by a box size of 512 pixel and then subjected to reference-free 2D classification. After particle screening using 2D and 3D classification, the final 355,289 particles for CsGSIb and 115,795 particles for GmGSβ2 were subjected to Ab-Initio Reconstitution and followed by 3D Refinement with C5 symmetry imposed. Four different conformational states were obtained for CsGSIb, resulting in a 3.3 Å density map for CsGSIb$^{Dec}$, 3.5 Å density map for CsGSIb$^{Pen}$ State I, 3.6 Å density map for CsGSIb$^{Pen}$ State II, and 3.4 Å density map for CsGSIb$^{Pen}$ State III. Only one major conformation was obtained with 2.9 Å density map for GmGSβ2. The global resolution of the map was estimated based on the gold-standard Fourier shell correlation (FSC) using the 0.143 criterion.

## Model building and structural refinement

Homology models of CsGSIb and GmGSβ2 were generated with the I-TASSER server (*Yang and Zhang, 2015*) and docked into the cryoEM maps using UCSF Chimera (*Pettersen et al., 2004*). The sequences were mutated with corresponding residues in CsGSIb and GmGSβ2, followed by rebuilding in Coot (*Emsley et al., 2010*). The missing residues of CsGSIb$^{Pen}$ were not built due to the lack of corresponding densities. Real-space refinement of models with geometry and secondary structure restraints applied was performed using PHENIX (*Afonine et al., 2018*). The final model was subjected to refinement and validation in PHENIX. The statistics of cryo-EM data collection, refinement and model validation are summarized in *Supplementary file 2*.

## Different nitrogen treatments for hydroponically grown tea cuttage seedlings

Two-year-old hydroponic tea cuttage seedlings were grown in a greenhouse at 20–25°C until new tender roots emerged. These healthy tea seedlings were then transferred into hydroponic solutions with different nitrogen sources, namely, 0 mM $NH_4^+$ (Shigeki Konishi solution), 5 mM $NH_4^+$ (Shigeki Konishi solution with 5 mM ammonium nitrogen), 10 mM $NH_4^+$ (Shigeki Konishi solution with 10 mM ammonium nitrogen), and a control (Shigeki Konishi solution alone). All these tea seedling root samples were cleaned and collected in liquid nitrogen after treatment for RNA analysis.

## RNA isolation and qRT-PCR analysis

Tea plant tissues or root materials were ground in liquid nitrogen into fine powders for total RNA extraction with an RNA extraction kit (Tiangen Biotech Co., Ltd.) according to the manufacturer's instructions. RNA quality and purity were assessed by a NanoDrop 2000 spectrophotometer (Thermo Scientific). The integrity of the RNA samples was rapidly checked by 1.0 % agarose gel electrophoresis. The total RNA was reverse-transcribed to single-stranded cDNAs using SuperScript III reverse

transcriptase (Invitrogen) according to the manufacturer's instructions. qRT-PCR analysis was performed using cDNA synthesized by the Prime Script RT Reagent Kit (Takara). Each qRT-PCR was conducted in a 20 μL reaction mixture containing 2 μL of diluted template cDNA, 0.4 μL of each specific primer, 10 μL of SYBR Premix Ex-Taq (Takara), and 7.2 μL of $H_2O$. All qRT-PCR assays were performed on the Bio-Rad CFX96 fluorescence-based quantitative PCR platform. The program used was as follows: 95 °C for 5 min; 40 cycles of 95 °C for 5 s for denaturation and 60 °C for 30 s for annealing and extension; and 61 cycles of 65 °C for 10 s for melting curve analysis. All experiments were independently repeated three times, and relative expression levels were measured using the $2^{-\Delta Ct}$ method.

## Subcellular localization of CsGSIs and Cs14-3-3-1a&1b

Construction of the Cs14-3-3-1a-GFP, CsGSI1b-GFP, and Cs14-3-3-1b-GFP fusions were performed using gateway recombination systems. The corresponding ORFs for CsGSIb and Cs14-3-3-1a, 1b were subcloned into pK7WGF2 in frame with a GFP tagged at the N-terminus. Determination of the subcellular localization of these GFP fusions was performed using tobacco leaf infiltration as previously described (*Zhao et al., 2011*). Briefly, the pK7WGF2-Cs14-3-3-1a-GFP, pK7WGF2-Cs14-3-3-1b-GFP, and pK7WGF2-CsGSIb-GFP plasmids were transformed into *A. tumefaciens* strain EHA105, and selected positive colonies harboring these constructs were used for plant transformation by infiltration. *Acetosyringone*-activated Agrobacterium cells were infiltrated into the *Nicotiana benthamiana* leaves leaf abaxial epidermal surface, and the tobacco plants were grown at room temperature for 3 days before imaging. Imaging of these GFP fusion proteins was performed using a confocal microscope with a 100× water immersion objective and appropriate software. The excitation wavelength was 488 nm, and emissions were collected at 500 nm.

## In planta interaction of CsGSIb with Cs14-3-3 proteins by bimolecular fluorescent complimentary (BIFC)

The ORFs of Cs14-3-3-1a, Cs14-3-3-1b, and CsGSIb were amplified and subcloned into pCAMBIA1300-eYFPN (YFP N-terminal portion) and pCAMBIA1300-eYFPC (YFP C-terminal portion) (CAMBIA) by the in-fusion technology. The resulting constructs were transformed into *A. tumefaciens* strains GV3101, which were infiltrated into Nicotiana benthamiana leaves individually or in different pair combinations. A Leica DMi8 M laser scanning confocal microscopy system was used for fluorescence observation, according to the method described previously (*Zhao et al., 2011*). If the fluorescence signal could be detected with any interaction pair, the pair of half YFP-fusion proteins should interact.

## Knockdown of Cs14-3-3-1a in tea plant hairy roots

Approximately 400 bp of the gene-specific fragments from Cs14-3-3-1a were amplified and subcloned into the final RNA interference (RNAi) destination vector pB7GWIWG by BP and LR clonase-based recombination reactions (Invitrogen). The resulting binary vectors pB7GWIWG-Cs14-3-3-1a were transformed into *A. rhizogenes* strain ATCC 15834 by electroporation. The selected positive transformants harboring pB7GWIWG-Cs14-3-3-1a were used to transform 3-month-old tea seedlings, which were pretreated with acetosyringone. The positive transgenic hairy root lines were verified with qRT-PCR for examination of transgene expression. At least three independent hairy root lines were selected for further analysis.

## Determination of free amino acids in tea plant samples

The free amino acids in tea plant samples were analyzed by using an amino acid analyzer (L-8900, Hitachi) according to manufacture instruction. The free amino acids were extracted from 120 mg leaves with 1 mL of 4 % sulfosalicylic acid in water bath sonication for 30 min and then centrifuged at 13,680 x g for 30 min. The debris was re-extracted once again and the supernatants from two extractions were combined as previously described (*Lu et al., 2019*). The supernatants were filtered through a 0.22 μm Millipore filter before analysis. A mobile phase containing lithium citrate for amino acid derivatization and UV–Vis detection at 570 and 440 nm were used in the Hitachi High-Speed Amino Acid Analyzer system. The flow rates were set at 0.35 mL/min for the mobile phase and 0.3 mL/min for the derivatization reagent. The temperature for separation column was set to 38 °C, and for the post-column reaction equipment was maintained at 130 °C. The temperature of the autosampler

was kept at 4 °C. The peak areas of amino acids were quantified in comparison with the amino acid standards.

## GS activity assay from plant samples

To determine the total GS activity, 100 mg of frozen plant samples were grounded into fine powder in liquid nitrogen. Samples were homogenized in extraction buffer (50 mmol/L Tris-HCl, pH 8.0, 2 mmol/L $MgSO_4$, 4 mmol/L dithiothreitol, and 0.4 mmol/L sucrose). Plant extracts were centrifuged at 13,680 x g (4 °C) for 25 min and the supernatants of extracts were analyzed for the soluble protein content using the Bradford assay. GS activity was determined after incubating the enzyme extracts in a reaction buffer (100 mmol/L Tris-HCl, 80 mmol/L $MgSO_4$, 20 mmol/L sodium glutamate, 80 mM $NH_4OH$, 20 mmol/L cysteine, 2 mmol/L EGTA and 40 mmol/L ATP) at 37 °C for 30 min (**Husted et al., 2002**). A stop solution containing 0.2 mol/L Trichloride acetic acid, 0.37 mol/L $FeCl_3$ and 0.6 mol/L HCl was added; and the absorbance of enzyme reactions at 540 nm was recorded. A standard curve was made in an identical way for calculation of the specific enzyme activity.

---

# Additional information

### Funding

| Funder | Grant reference number | Author |
|---|---|---|
| National Natural Science Foundation of China | 31770807 | Chengdong Huang |
| National Natural Science Foundation of China | 31971144 | Chengdong Huang |
| The Open Fund of State Key Laboratory of Tea Plant Biology and Utilization | SKLTOF20190102 | Chengdong Huang |
| National Key Research and Development Program of China | Project in Green Bio-production 2021YFC2100100 | Qiong Xing |
| National Key Research and Development Program of China | 2018YFD1000601 | Jian Zhao |
| Key Research and Development (R&D) Program of Anhui Province | 18030701155 | Jian Zhao |

The funders had no role in study design, data collection and interpretation, or the decision to submit the work for publication.

### Author contributions

Yao Chen, Weiya Xu, Shuwei Yu, Data curation, Formal analysis; Kang Ni, Guangbiao She, Data curation; Xiaodong Ye, Qiong Xing, Formal analysis; Jian Zhao, Formal analysis, Investigation, Supervision; Chengdong Huang, Conceptualization, Supervision, Writing - original draft, Writing - review and editing

### Author ORCIDs

Weiya Xu http://orcid.org/0000-0001-5796-8758
Qiong Xing http://orcid.org/0000-0003-0213-8364
Jian Zhao http://orcid.org/0000-0002-4416-7334
Chengdong Huang http://orcid.org/0000-0002-8997-9459

### Decision letter and Author response

Decision letter https://doi.org/10.7554/eLife.72535.sa1
Author response https://doi.org/10.7554/eLife.72535.sa2

## Additional files

### Supplementary files

• Supplementary file 1. Structural comparisons of GmGSβ2 and CsGSIb$^{Dec}$ with glutamine synthetase from various species.
• Supplementary file 2. Cryo-EM data collection, refinement, and validation statistics.
• Transparent reporting form

### Data availability

Structures generated in this study have been deposited in PDB under the accession code 7V4H, 7V4I, 7V4J, 7V4K, and 7V4L. All data generated or analysed during this study are included in the manuscript and supporting file.

The following dataset was generated:

| Author(s) | Year | Dataset title | Dataset URL | Database and Identifier |
| --- | --- | --- | --- | --- |
| Xu W, Chen Y, Xing Q, Huang C | 2021 | Cryo-EM Structure of Glycine max glutamine synthetase GmGS Beta2 | https://www.rcsb.org/structure/7V4H | RCSB Protein Data Bank, 7V4H |
| Xu W, Chen Y, Xing Q, Huang C | 2021 | Cryo-EM Structure of Camellia sinensis glutamine synthetase CsGSIb decamer assembly | https://www.rcsb.org/structure/7V4I | RCSB Protein Data Bank, 7V4I |
| Xu W, Chen Y, Xing Q, Huang C | 2021 | Cryo-EM Structure of Camellia sinensis glutamine synthetase CsGSIb inactive Pentamer State I | https://www.rcsb.org/structure/7V4J | RCSB Protein Data Bank, 7V4J |
| Xu W, Chen Y, Xing Q, Huang C | 2021 | Cryo-EM Structure of Camellia sinensis glutamine synthetase CsGSIb inactive Pentamer State II | https://www.rcsb.org/structure/7V4K | RCSB Protein Data Bank, 7V4K |
| Xu W, Chen Y, Xing Q, Huang C | 2021 | Cryo-EM Structure of Camellia sinensis glutamine synthetase CsGSIb inactive Pentamer State III | https://www.rcsb.org/structure/7V4L | RCSB Protein Data Bank, 7V4L |

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
