## [Editor Report]

This study provides deep structural insights into how an important enzyme, glutamine synthase, which exists in at least two large multimeric complexes, can be catalytically regulated in different assemblies. The paper combines high resolution cryoEM, enzymology, and cellular studies to provide a plausible model for how the intricate structural changes regulate activity.

---

## [Decision Letter]

**Decision letter after peer review:**

Thank you for submitting your article "Assembly status transition offers an avenue for activity modulation of a supramolecular enzyme via dynamics-driven allostery" for consideration by *eLife*. Your article has been reviewed by 2 peer reviewers, and the evaluation has been overseen by a Reviewing Editor and Philip Cole as the Senior Editor. The following individual involved in review of your submission has agreed to reveal their identity: Xin Zhang (Reviewer #1).

Essential revisions:

1) The authors made mutations to the ring-ring interface region of CsGSIb, which converts this enzyme from a weakly decamer-forming enzyme to a stable decamer, and observed a drastic increase in activity. We suggest the authors also make mutations to GmGSβ2 to disrupt its decamer assembly, and evaluate mutational effects on the enzymatic activity.

2) A few GS structures from various species have been reported. The authors should compare the structures solved in this paper with others and provide information how GS structure is conserved across species.

3) In the figure legend of Figure 1, GmGSβ should be corrected to GmGSβ2.

4) Nomenclature of enzymes needs to be consistent: is it CsGSIb or CsGS1-Ib?

5) A major weakness is the presentation and interpretation of the experimental results. We don't see the need to invoke the "dynamics-driven assembly" or "allosteric modulation", and we believe these terms are imprecise here. The rim regions in the pentamers are flexible and become ordered when pentamers dimerize. But the dynamism per se does not drive the assembly. An unfolded protein is dynamics, and upon folding, the catalytic pocket is formed, and the protein becomes active. But one cannot say the folding process is "driven" by dynamism. Other evidence arguing against the "dynamics-driven assembly" is the authors' own experiment in which the dimerization of the weak binding pentamers is mediated by 14-3-3 chaperone, not driven by the dynamics of the pentamers.

6) We don't see the allosteric paradigm. During dimerization, the flexible rim region becomes ordered, and the catalytic pocket resides in the disordered rim region. This is a classic binding-induced folding and activation process. In fact, the authors' own observation argues against the so-called allostery: in the pentamer, the ordered region is the same as in the decamer, so there is only binding induced disorder-to order transition and there is no allostery.

7) Given changes suggested in 5 and 6 the authors focus on their immediate experimental insights and remove those grandiose terms and unnecessary (perhaps inappropriate) generalization.

8) The main text seems terse and should be tightened up.

*Reviewer #1:*

Strengths:

The authors combined a series of biochemical and biophysical investigations and showed compelling evidence that the activity of GS is remotely modulated by the contacts between two oligomeric rings.

The authors further solved the cryo-EM structures of the GS in both the catalytically active states (two oligomeric rings attached) and inactive assembly states (isolated pentamer rings), which allow them to propose a "dynamics-driven" allostery mechanism.

The authors demonstrated in vivo evidence that the activity of CsGSIb is activated by a universal scaffold protein, the 14-3-3 protein. As it has been established that the 14-3-3 protein could modulate protein activity through dimerization, the allostery mechanism presented in this paper provides a plausible explanation that how eukaryotic GS is activated by the 14-3-3 protein.

Weakness and suggestions:

The authors made mutations to the ring-ring interface region of CsGSIb, which converts this enzyme from a weakly decamer-forming enzyme to a stable decamer, and observed a drastic increase in activity. I suggest the authors also make mutations to GmGSβ2 to disrupt its decamer assembly, and evaluate mutational effects on the enzymatic activity.

A few GS structures from various species have been reported. The authors should compare the structures solved in this paper with others and provide information how GS structure is conserved across species.

In the figure legend of Figure 1, GmGSβ should be corrected to GmGSβ2.

Nomenclature of enzymes needs to be consistent: is it CsGSIb or CsGS1-Ib?

*Reviewer #2:*

This is a comparative study of two plant glutamine synthetases. The authors show that one enzyme is less active in vitro, due to the equilibrium between the inactive pentamer and the active dimer-of-pentamer forms, and the other is a stable dimer-of-pentamer decamer and is highly active. Through structural analysis, they show – for the first time – that the rim regions of the pentameric – where the active site residues – are flexible, explaining the low activity of the pentamer and the need to assemble the dimer-of-pentamers. This molecular level insight is complemented with a cellular study in which they show that the weakly-binding pentamers require the scaffolding protein 14-3-3 in vivo, which mediates dimerization of the GS pentamers. The structural analysis is solid. The work does improve our understanding of the GS, an important enzyme in plant as well as in other organisms.

A major weakness is the presentation and interpretation of the experimental results. I don't see the need to invoke the "dynamics-driven assembly" or "allosteric modulation", and I believe these terms are used inaccurately here. The rim regions in the pentamers are flexible and become ordered when pentamers dimerize. But the dynamism per se does not drive the assembly. An unfolded protein is dynamics, and upon folding, the catalytic pocket is formed, and the protein becomes active. But one cannot say the folding process is "driven" by dynamism. Another evidence arguing against the "dynamics-driven assembly" is the authors' own experiment in which the dimerization of the weak binding pentamers is mediated by 14-3-3 chaperone, not driven by the dynamics of the pentamers

By the same token, I don't see the allosteric paradigm. During dimerization, the flexible rim region become ordered, and the catalytic pocket resides in the disordered rim region. This is a classic binding-induced folding and activation process. In fact, the authors' own observation argues against the so-called allostery: in the pentamer, the ordered region is the same as in the decamer, so there is only binding induced disorder-to order transition and there is no allostery.

I suggest that the authors focus on their immediate experimental insights and remove those grandiose terms and unnecessary (perhaps inappropriate) generalization. The main text seems terse and should be tightened up.

---

## [Author Response]

Essential revisions:1) The authors made mutations to the ring-ring interface region of CsGSIb, which converts this enzyme from a weakly decamer-forming enzyme to a stable decamer, and observed a drastic increase in activity. We suggest the authors also make mutations to GmGSβ2 to disrupt its decamer assembly, and evaluate mutational effects on the enzymatic activity.

We totally agree with the reviewer’s suggestion. In fact, we did make mutations to GmGSβ2. Unfortunately, we found the *GmGSβ2* mutants were expressed as inclusion body and we failed to isolate any soluble protein sample even we tried refolding.

2) A few GS structures from various species have been reported. The authors should compare the structures solved in this paper with others and provide information how GS structure is conserved across species.

We have added the Supplementary File 1 to provide structural comparison information of CsGSIb and GmGSβ2 with GSs from some representative species and added structural comparison information in the text accordingly: “Indeed, structural alignments of GSs reveals a conservative catalytic site geometry across a wide range of species (Supplementary File 1).”

3) In the figure legend of Figure 1, GmGSβ should be corrected to GmGSβ2.

We have corrected this. Thanks for picking it up.

4) Nomenclature of enzymes needs to be consistent: is it CsGSIb or CsGS1-Ib?

It should be CsGSIb. We have corrected this issue.

5) A major weakness is the presentation and interpretation of the experimental results. We don't see the need to invoke the "dynamics-driven assembly" or "allosteric modulation", and we believe these terms are imprecise here. The rim regions in the pentamers are flexible and become ordered when pentamers dimerize. But the dynamism per se does not drive the assembly. An unfolded protein is dynamics, and upon folding, the catalytic pocket is formed, and the protein becomes active. But one cannot say the folding process is "driven" by dynamism. Other evidence arguing against the "dynamics-driven assembly" is the authors' own experiment in which the dimerization of the weak binding pentamers is mediated by 14-3-3 chaperone, not driven by the dynamics of the pentamers.

We think it over and agree with the reviewer’s opinion. We have removed the descriptions of “dynamics-driven allostery” or “allosteric modulation”, and replaced such descriptions with “binding induced disorder-order transition” throughout the entire manuscript. We have modified the title of this manuscript as well.

6) We don't see the allosteric paradigm. During dimerization, the flexible rim region becomes ordered, and the catalytic pocket resides in the disordered rim region. This is a classic binding-induced folding and activation process. In fact, the authors' own observation argues against the so-called allostery: in the pentamer, the ordered region is the same as in the decamer, so there is only binding induced disorder-to order transition and there is no allostery.

We have modified the text as well as the Figures (Figure 4 and Figure 6) to remove all description of allostery, according to the reviewer’s suggestion.

7) Given changes suggested in 5 and 6 the authors focus on their immediate experimental insights and remove those grandiose terms and unnecessary (perhaps inappropriate) generalization.

We have removed the entire part of “Dynamics-driven allostery induced by assembly status transition” in Discussion and modified the manuscript according to the reviewer’s suggestion.

8) The main text seems terse and should be tightened up.

We have tightened the manuscript and removed ~500 words in total.